# Spectrally-normalized margin bounds
# for neural networks

**Peter L. Bartlett**[*]  **Dylan J. Foster**[†]  **Matus Telgarsky**[‡]

## Abstract

This paper presents a margin-based multiclass generalization bound for neural networks that scales with their margin-normalized *spectral complexity*: their Lipschitz constant, meaning the product of the spectral norms of the weight matrices, times a certain correction factor. This bound is empirically investigated for a standard AlexNet network trained with SGD on the `mnist` and `cifar10` datasets, with both original and random labels; the bound, the Lipschitz constants, and the excess risks are all in direct correlation, suggesting both that SGD selects predictors whose complexity scales with the difficulty of the learning task, and secondly that the presented bound is sensitive to this complexity.

## 1 Overview

Neural networks owe their astonishing success not only to their ability to fit any data set: they also *generalize well*, meaning they provide a close fit on unseen data. A classical statistical adage is that models capable of fitting too much will generalize poorly; what's going on here?

Let's navigate the many possible explanations provided by statistical theory. A first observation is that any analysis based solely on the number of possible labellings on a finite training set — as is the case with VC dimension — is doomed: if the function class can fit all possible labels (as is the case with neural networks in standard configurations [Zhang et al., 2017]), then this analysis can not distinguish it from the collection of all possible functions!

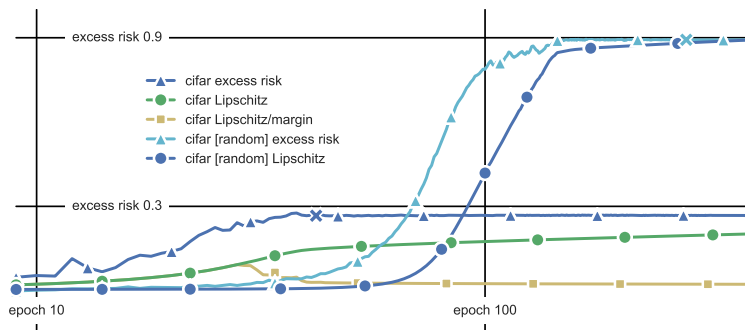

Figure 1: An analysis of AlexNet [Krizhevsky et al., 2012] trained with SGD on `cifar10`, both with original and with random labels. Triangle-marked curves track excess risk across training epochs (on a log scale), with an 'x' marking the earliest epoch with zero training error. Circle-marked curves track Lipschitz constants, normalized so that the two curves for random labels meet. The Lipschitz constants tightly correlate with excess risk, and moreover normalizing them by *margins* (resulting in the square-marked curve) neutralizes growth across epochs.

[*]`<peter@berkeley.edu>`; University of California, Berkeley and Queensland University of Technology.
[†]`<djf244@cornell.edu>`; Cornell University.
[‡]`<mjt@illinois.edu>`; University of Illinois, Urbana-Champaign.

Next let's consider *scale-sensitive* measures of complexity, such as Rademacher complexity and metric entropy, which work directly with real-valued function classes, and moreover are sensitive to their magnitudes. Figure 1 plots the excess risk (the test error minus the training error) across training epochs against one candidate scale-sensitive complexity measure, the Lipschitz constant of the network (the product of the spectral norms of their weight matrices), and demonstrates that they are tightly correlated (which is not the case for, say, the $l_2$ norm of the weights). The data considered in Figure 1 is the standard `cifar10` dataset, both with original and with random labels, which has been used as a sanity check when investigating neural network generalization [Zhang et al., 2017].

There is still an issue with basing a complexity measure purely on the Lipschitz constant (although it has already been successfully employed to regularize neural networks [Cisse et al., 2017]): as depicted in Figure 1, the measure grows over time, despite the excess risk plateauing. Fortunately, there is a standard resolution to this issue: investigating the *margins* (a precise measure of confidence) of the outputs of the network. This tool has been used to study the behavior of 2-layer networks, boosting methods, SVMs, and many others [Bartlett, 1996, Schapire et al., 1997, Boucheron et al., 2005]; in boosting, for instance, there is a similar growth in complexity over time (each training iteration adds a weak learner), whereas margin bounds correctly stay flat or even decrease. This behavior is recovered here: as depicted in Figure 1, even though standard networks exhibit growing Lipschitz constants, normalizing these Lipschitz constants by the margin instead gives a decaying curve.

## 1.1 Contributions

This work investigates a complexity measure for neural networks that is based on the Lipschitz constant, but normalized by the margin of the predictor. The two central contributions are as follows.

- Theorem 1.1 below will give the rigorous statement of the generalization bound that is the basis of this work. In contrast to prior work, this bound: **(a)** scales with the Lipschitz constant (product of spectral norms of weight matrices) divided by the margin; **(b)** has no dependence on combinatorial parameters (e.g., number of layers or nodes) outside of log factors; **(c)** is multiclass (with no explicit dependence on the number of classes); **(d)** measures complexity against a *reference network* (e.g., for the ResNet [He et al., 2016], the reference network has identity mappings at each layer). The bound is stated below, with a general form and analysis summary appearing in Section 3 and the full details relegated to the appendix.

- An empirical investigation, in Section 2, of neural network generalization on the standard datasets `cifar10`, `cifar100`, and `mnist` using the preceding bound. Rather than using the bound to provide a single number, it can be used to form a *margin distribution* as in Figure 2. These margin distributions will illuminate the following intuitive observations: **(a)** `cifar10` is harder than `mnist`; **(b)** random labels make `cifar10` and `mnist` much more difficult; **(c)** the margin distributions (and bounds) converge during training, even though the weight matrices continue to grow; **(d)** $l_2$ regularization ("weight decay") does not significantly impact margins or generalization.

A more detailed description of the margin distributions is as follows. Suppose a neural network computes a function $f : \mathbb{R}^d \to \mathbb{R}^k$, where $k$ is the number of classes; the most natural way to convert this to a classifier is to select the output coordinate with the largest magnitude, meaning $x \mapsto \arg\max_j f(x)_j$. The *margin*, then, measures the gap between the output for the correct label and other labels, meaning $f(x)_y - \max_{j \neq y} f(x)_j$.

Unfortunately, margins alone do not seem to say much; see for instance Figure 2a, where the collections of all margins for all data points — the *unnormalized margin distribution* — are similar for `cifar10` with and without random labels. What is missing is an appropriate *normalization*, as in Figure 2b. This normalization is provided by Theorem 1.1, which can now be explained in detail.

To state the bound, a little bit of notation is necessary. The networks will use $L$ fixed nonlinearities $(\sigma_1, \ldots, \sigma_L)$, where $\sigma_i : \mathbb{R}^{d_{i-1}} \to \mathbb{R}^{d_i}$ is $\rho_i$-Lipschitz (e.g., as with coordinate-wise ReLU, and max-pooling, as discussed in Appendix A.1); occasionally, it will also hold that $\sigma_i(0) = 0$. Given $L$ weight matrices $\mathcal{A} = (A_1, \ldots, A_L)$ let $F_{\mathcal{A}}$ denote the function computed by the corresponding network:

$$F_{\mathcal{A}}(x) := \sigma_L(A_L \sigma_{L-1}(A_{L-1} \cdots \sigma_1(A_1 x) \cdots)). \tag{1.1}$$

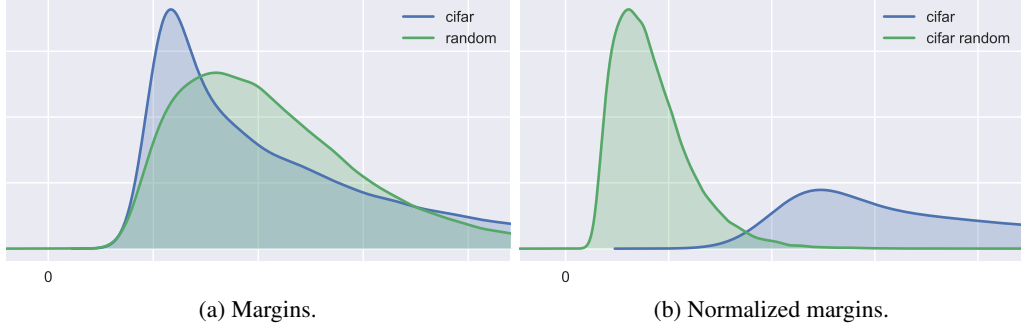

(a) Margins.                    (b) Normalized margins.

Figure 2: Margin distributions at the end of training AlexNet on `cifar10`, with and without random labels. With proper normalization, random labels demonstrably correspond to a harder problem.

The network output $F_{\mathcal{A}}(x) \in \mathbb{R}^{d_L}$ (with $d_0 = d$ and $d_L = k$) is converted to a class label in $\{1, \ldots, k\}$ by taking the $\arg\max$ over components, with an arbitrary rule for breaking ties. Whenever input data $x_1, \ldots, x_n \in \mathbb{R}^d$ are given, collect them as rows of a matrix $X \in \mathbb{R}^{n \times d}$. Occasionally, notation will be overloaded to discuss $F_{\mathcal{A}}(X^T)$, a matrix whose $i^{\text{th}}$ column is $F_{\mathcal{A}}(x_i)$. Let $W$ denote the maximum of $\{d, d_1, \ldots, d_L\}$. The $l_2$ norm $\|\cdot\|_2$ is always computed entry-wise; thus, for a matrix, it corresponds to the Frobenius norm.

Next, define a collection of *reference matrices* $(M_1, \ldots, M_L)$ with the same dimensions as $A_1, \ldots, A_L$; for instance, to obtain a good bound for ResNet [He et al., 2016], it is sensible to set $M_i := I$, the identity map, and the bound below will worsen as the network moves farther from the identity map; for AlexNet [Krizhevsky et al., 2012], the simple choice $M_i = 0$ suffices. Finally, let $\|\cdot\|_\sigma$ denote the spectral norm and $\|\cdot\|_{p,q}$ denote the $(p, q)$ matrix norm, defined by $\|A\|_{p,q} := \left\| (\|A_{:,1}\|_p, \ldots, \|A_{:,m}\|_p) \right\|_q$ for $A \in \mathbb{R}^{d \times m}$. The *spectral complexity* $R_{F_{\mathcal{A}}} = R_{\mathcal{A}}$ of a network $F_{\mathcal{A}}$ with weights $\mathcal{A}$ is the defined as

$$R_{\mathcal{A}} := \left( \prod_{i=1}^{L} \rho_i \|A_i\|_\sigma \right) \left( \sum_{i=1}^{L} \frac{\|A_i^\top - M_i^\top\|_{2,1}^{2/3}}{\|A_i\|_\sigma^{2/3}} \right)^{3/2}. \tag{1.2}$$

The following theorem provides a generalization bound for neural networks whose nonlinearities are fixed but whose weight matrices $\mathcal{A}$ have bounded spectral complexity $R_{\mathcal{A}}$.

**Theorem 1.1.** *Let nonlinearities $(\sigma_1, \ldots, \sigma_L)$ and reference matrices $(M_1, \ldots, M_L)$ be given as above (i.e., $\sigma_i$ is $\rho_i$-Lipschitz and $\sigma_i(0) = 0$). Then for $(x, y), (x_1, y_1), \ldots, (x_n, y_n)$ drawn iid from any probability distribution over $\mathbb{R}^d \times \{1, \ldots, k\}$, with probability at least $1 - \delta$ over $((x_i, y_i))_{i=1}^n$, every margin $\gamma > 0$ and network $F_{\mathcal{A}} : \mathbb{R}^d \to \mathbb{R}^k$ with weight matrices $\mathcal{A} = (A_1, \ldots, A_L)$ satisfy*

$$\Pr\left[ \arg\max_j F_{\mathcal{A}}(x)_j \neq y \right] \leq \widehat{\mathcal{R}}_\gamma(F_{\mathcal{A}}) + \widetilde{\mathcal{O}}\left( \frac{\|X\|_2 R_{\mathcal{A}}}{\gamma n} \ln(W) + \sqrt{\frac{\ln(1/\delta)}{n}} \right),$$

*where $\widehat{\mathcal{R}}_\gamma(f) \leq n^{-1} \sum_i \mathbb{1}\left[ f(x_i)_{y_i} \leq \gamma + \max_{j \neq y_i} f(x_i)_j \right]$ and $\|X\|_2 = \sqrt{\sum_i \|x_i\|_2^2}$.*

The full proof and a generalization beyond spectral norms is relegated to the appendix, but a sketch is provided in Section 3, along with a lower bound. Section 3 also gives a discussion of related work: briefly, it's essential to note that margin and Lipschitz-sensitive bounds have a long history in the neural networks literature [Bartlett, 1996, Anthony and Bartlett, 1999, Neyshabur et al., 2015]; the distinction here is the sensitivity to the spectral norm, and that there is no explicit appearance of combinatorial quantities such as numbers of parameters or layers (outside of log terms, and indices to summations and products).

To close, miscellaneous observations and open problems are collected in Section 4.

## 2 Generalization case studies via margin distributions

In this section, we empirically study the generalization behavior of neural networks, via margin distributions and the generalization bound stated in Theorem 1.1.

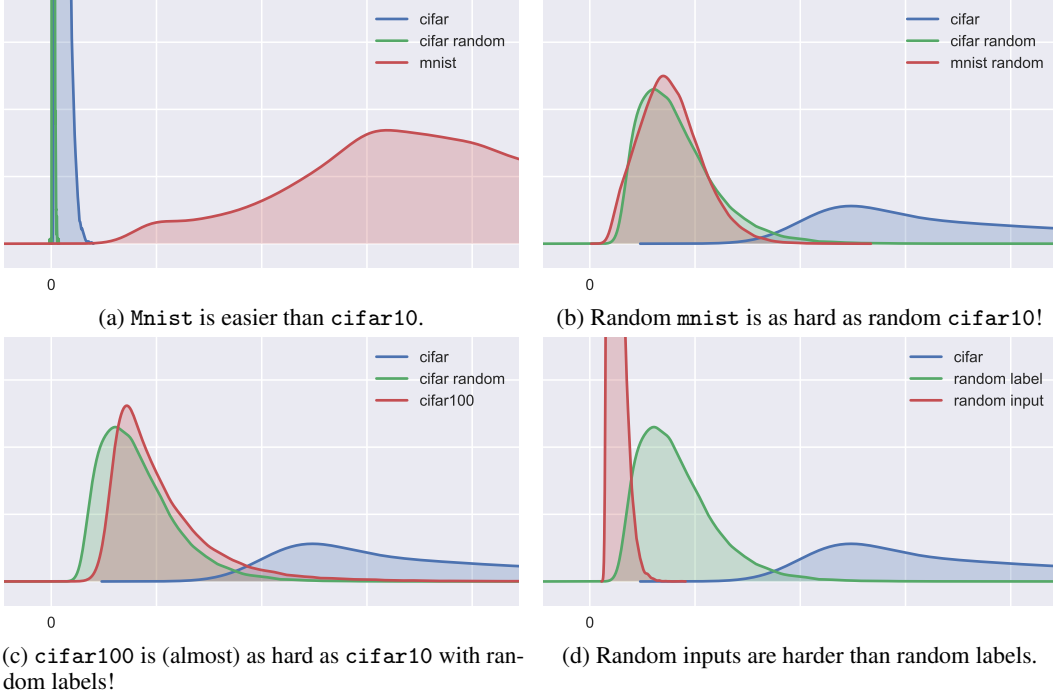

(a) Mnist is easier than `cifar10`.

(b) Random `mnist` is as hard as random `cifar10`!

(c) `cifar100` is (almost) as hard as `cifar10` with random labels!

(d) Random inputs are harder than random labels.

Figure 3: A variety of margin distributions. Axes are re-scaled in Figure 3a, but identical in the other subplots; the `cifar10` (blue) and random `cifar10` (green) distributions are the same each time.

Before proceeding with the plots, it's a good time to give a more refined description of the margin distribution, one that is suitable for comparisons across datasets. Given $n$ pattern/label pairs $((x_i, y_i))_{i=1}^{n}$, with patterns as rows of matrix $X \in \mathbb{R}^{n \times d}$, and given a predictor $F_{\mathcal{A}} : \mathbb{R}^d \to \mathbb{R}^k$, the (normalized) margin distribution is the univariate empirical distribution of the labeled data points each transformed into a single scalar according to

$$(x, y) \mapsto \frac{F_{\mathcal{A}}(x)_y - \max_{i \neq y} F_{\mathcal{A}}(x)_i}{R_{\mathcal{A}} \|X\|_2 / n},$$

where the spectral complexity $R_{\mathcal{A}}$ is from eq. (1.2). The normalization is thus derived from the bound in Theorem 1.1, but ignoring log terms.

Taken this way, the two margin distributions for two datasets can be interpreted as follows. Considering any fixed point on the horizontal axis, if the *cumulative* distribution of one density is lower than the other, then it corresponds to a lower right hand side in Theorem 1.1. For no reason other than visual interpretability, the plots here will instead depict a density estimate of the margin distribution. The vertical and horizontal axes are rescaled in different plots, but the random and true `cifar10` margin distributions are always the same.

A little more detail about the experimental setup is as follows. All experiments were implemented in Keras [Chollet et al., 2015]. In order to minimize conflating effects of optimization and regularization, the optimization method was vanilla SGD with step size $0.01$, and all regularization (weight decay, batch normalization, etc.) were disabled. "`cifar`" in general refers to `cifar10`, however `cifar100` will also be explicitly mentioned. The network architecture is essentially AlexNet [Krizhevsky et al., 2012] with all normalization/regularization removed, and with no adjustments of any kind (even to the learning rate) across the different experiments.

**Comparing datasets.** A first comparison is of `cifar10` and the standard `mnist` digit data. `mnist` is considered "easy", since any of a variety of methods can achieve roughly 1% test error. The "easiness" is corroborated by Figure 3a, where the margin distribution for `mnist` places all its mass far to the right of the mass for `cifar10`. Interestingly, randomizing the labels of `mnist`, as in Figure 3b, results in a margin distribution to the left of not only `cifar10`, but also slightly to the left of (but close to) `cifar10` with randomized labels.

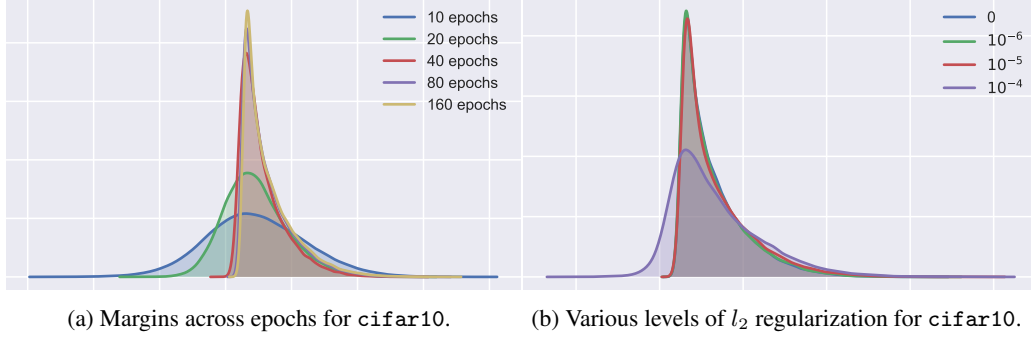

(a) Margins across epochs for `cifar10`.    (b) Various levels of $l_2$ regularization for `cifar10`.

Figure 4

Next, Figure 3c compares `cifar10` and `cifar100`, where `cifar100` uses the same input images as `cifar10`; indeed, `cifar10` is obtained from `cifar100` by collapsing the original 100 categories into 10 groups. Interestingly, `cifar100`, from the perspective of margin bounds, is just as difficult as `cifar10` with random labels. This is consistent with the large observed test error on `cifar100` (which has not been "optimized" in any way via regularization).

Lastly, Figure 3d replaces the `cifar10` *input images* with random images sampled from Gaussians matching the first- and second-order image statistics (see [Zhang et al., 2017] for similar experiments).

**Convergence of margins.**    As was pointed out in Section 1, the weights of the neural networks do not seem to converge in the usual sense during training (the norms grow continually). However, as depicted in Figure 4a, the sequence of (normalized) margin distributions is itself converging.

**Regularization.**    As remarked in [Zhang et al., 2017], regularization only seems to bring minor benefits to test error (though adequate to be employed in all cutting edge results). This observation is certainly consistent with the margin distributions in Figure 4b, which do not improve (e.g., by shifting to the right) in any visible way under regularization. An open question, discussed further in Section 4, is to design regularization that improves margins.

# 3    Analysis of margin bound

This section will sketch the proof of Theorem 1.1, give a lower bound, and discuss related work.

## 3.1    Multiclass margin bound

The starting point of this analysis is a margin-based bound for multiclass prediction. To state the bound, first recall that the *margin operator* $\mathcal{M} : \mathbb{R}^k \times \{1, \ldots, k\} \to \mathbb{R}$ is defined as $\mathcal{M}(v, y) := v_y - \max_{i \neq y} v_i$, and define the *ramp loss* $\ell_\gamma : \mathbb{R} \to \mathbb{R}^+$ as

$$\ell_\gamma(r) := \begin{cases} 0 & r < -\gamma, \\ 1 + r/\gamma & r \in [-\gamma, 0], \\ 1 & r > 0, \end{cases}$$

and *ramp risk* as $\mathcal{R}_\gamma(f) := \mathbb{E}(\ell_\gamma(-\mathcal{M}(f(x), y)))$. Given a sample $S := ((x_1, y_1), \ldots, (x_n, y_n))$, define an empirical counterpart $\widehat{\mathcal{R}}_\gamma$ of $\mathcal{R}_\gamma$ as $\widehat{\mathcal{R}}_\gamma(f) := n^{-1} \sum_i \ell_\gamma(-\mathcal{M}(f(x_i), y_i))$; note that $\mathcal{R}_\gamma$ and $\widehat{\mathcal{R}}_\gamma$ respectively upper bound the probability and fraction of errors on the source distribution and training set. Lastly, given a set of real-valued functions $\mathcal{H}$, define the *Rademacher complexity* as $\mathfrak{R}(\mathcal{H}_{|S}) := n^{-1} \mathbb{E} \sup_{h \in \mathcal{H}} \sum_{i=1}^n \epsilon_i h(x_i, y_i)$, where the expectation is over the Rademacher random variables $(\epsilon_1, \ldots, \epsilon_n)$, which are independent, uniform $\pm 1$-valued.

With this notation in place, the basic bound is as follows.

**Lemma 3.1.**    *Given functions $\mathcal{F}$ with $\mathcal{F} \ni f : \mathbb{R}^d \to \mathbb{R}^k$ and any $\gamma > 0$, define $\mathcal{F}_\gamma := \{(x, y) \mapsto \ell_\gamma(-\mathcal{M}(f(x), y)) : f \in \mathcal{F}\}$. Then, with probability at least $1 - \delta$ over a sample $S$ of size $n$, every $f \in \mathcal{F}$ satisfies $\Pr[\arg\max_i f(x)_i \neq y] \leq \widehat{\mathcal{R}}_\gamma(f) + 2\mathfrak{R}((\mathcal{F}_\gamma)_{|S}) + 3\sqrt{\frac{\ln(1/\delta)}{2n}}$.*

This bound is a direct consequence of standard tools in Rademacher complexity. In order to instantiate this bound, covering numbers will be used to directly upper bound the Rademacher complexity term

$\mathfrak{R}((\mathcal{F}_\gamma)_{|S})$. Interestingly, the choice of directly working in terms of covering numbers seems essential to providing a bound with no explicit dependence on $k$; by contrast, prior work primarily handles multiclass via a Rademacher complexity analysis on each coordinate of a $k$-tuple of functions, and pays a factor of $\sqrt{k}$ [Zhang, 2004].

## 3.2 Covering number complexity upper bounds

This subsection proves Theorem 1.1 via Lemma 3.1 by controlling, via covering numbers, the Rademacher complexity $\mathfrak{R}((\mathcal{F}_\gamma)_{|S})$ for networks with bounded spectral complexity.

The notation here for (proper) covering numbers is as follows. Let $\mathcal{N}(U, \epsilon, \|\cdot\|)$ denote the least cardinality of any subset $V \subseteq U$ that *covers* $U$ at scale $\epsilon$ with norm $\|\cdot\|$, meaning

$$\sup_{A \in U} \min_{B \in V} \|A - B\| \le \epsilon.$$

Choices of $U$ that will be used in the present work include both the image $\mathcal{F}_{|S}$ of data $S$ under some function class $F$, as well as the conceptually simpler choice of a family of matrix products.

The full proof has the following steps: **(I)** A *matrix covering* bound for the affine transformation of each layer is provided in Lemma 3.2; handling whole layers at once allows for more flexible norms. **(II)** An induction on layers then gives a covering number bound for entire networks; this analysis is only sketched here for the special case of norms used in Theorem 1.1, but the full proof in the appendix culminates in a bound for more general norms (cf. Lemma A.7). **(III)** The preceding whole-network covering number leads to Theorem 1.1 via Lemma 3.1 and standard techniques.

Step **(I)**, *matrix covering*, is handled by the following lemma. The covering number considers the matrix product $XA$, where $A$ will be instantiated as the weight matrix for a layer, and $X$ is the data passed through all layers prior to the present layer.

**Lemma 3.2.** *Let conjugate exponents $(p, q)$ and $(r, s)$ be given with $p \le 2$, as well as positive reals $(a, b, \epsilon)$ and positive integer $m$. Let matrix $X \in \mathbb{R}^{n \times d}$ be given with $\|X\|_p \le b$. Then*

$$\ln \mathcal{N}\left(\left\{XA : A \in \mathbb{R}^{d \times m}, \|A\|_{q,s} \le a\right\}, \epsilon, \|\cdot\|_2\right) \le \left\lceil \frac{a^2 b^2 m^{2/r}}{\epsilon^2} \right\rceil \ln(2dm).$$

The proof relies upon the *Maurey sparsification lemma* [Pisier, 1980], which is stated in terms of sparsifying convex hulls, and in its use here is inspired by covering number bounds for linear predictors [Zhang, 2002]. To prove Theorem 1.1, this matrix covering bound will be instantiated for the case of $\|A\|_{2,1}$. It is possible to instead scale with $\|A\|_2$ and $\|X\|_2$, but even for the case of the identity matrix $X = I$, this incurs an extra dimension factor. The use of $\|A\|_{2,1}$ here thus helps Theorem 1.1 avoid any appearance of $W$ and $L$ outside of log terms; indeed, the goal of covering a whole matrix at a time (rather than the more standard vector covering) was to allow this greater sensitivity and avoid combinatorial parameters.

Step **(II)** above, the induction on layers, proceeds as follows. Let $X_i$ denote the output of layer $i$ (thus $X_0 = X$), and inductively suppose there exists a cover element $\widehat{X}_i$ depending on covering matrices $(\widehat{A}_1, \ldots, \widehat{A}_{i-1})$ chosen to cover weight matrices in earlier layers. Thanks to Lemma 3.2, there also exists $\widehat{A}_i$ so that $\|A_i \widehat{X}_i - \widehat{A}_i \widehat{X}_i\|_2 \le \epsilon_i$. The desired cover element is thus $\widehat{X}_{i+1} = \sigma_i(\widehat{A}_i \widehat{X}_i)$ where $\sigma_i$ is the nonlinearity in layer $i$; indeed, supposing $\sigma_i$ is $\rho_i$-Lipschitz,

$$\begin{aligned}
\|X_{i+1} - \widehat{X}_{i+1}\|_2 &\le \rho_i \|A_i X_i - \widehat{A}_i \widehat{X}_i\|_2 \\
&\le \rho_i \left(\|A_i X_i - A_i \widehat{X}_i\|_2 + \|A_i \widehat{X}_i - \widehat{A}_i \widehat{X}_i\|_2\right) \\
&\le \rho_i \|A_i\|_\sigma \|X_i - A_i \widehat{X}_i\|_2 \rho_i + \epsilon_i,
\end{aligned}$$

where the first term is controlled with the inductive hypothesis. Since $\widehat{X}_{i+1}$ depends on each choice $(\widehat{A}_i, \ldots, \widehat{A}_i)$, the cardinality of the full network cover is the product of the individual matrix covers.

The preceding proof had no sensitivity to the particular choice of norms; it merely required an operator norm on $A_i$, as well as some other norm that allows matrix covering. Such an analysis is presented in full generality in Appendix A.5. Specializing to the particular case of spectral norms and $(2, 1)$ group norms leads to the following full-network covering bound.

**Theorem 3.3.** *Let fixed nonlinearities* $(\sigma_1, \ldots, \sigma_L)$ *and reference matrices* $(M_1, \ldots, M_L)$ *be given, where* $\sigma_i$ *is* $\rho_i$*-Lipschitz and* $\sigma_i(0) = 0$*. Let spectral norm bounds* $(s_1, \ldots, s_L)$*, and matrix* $(2, 1)$ *norm bounds* $(b_1, \ldots, b_L)$ *be given. Let data matrix* $X \in \mathbb{R}^{n \times d}$ *be given, where the* $n$ *rows correspond to data points. Let* $\mathcal{H}_X$ *denote the family of matrices obtained by evaluating* $X$ *with all choices of network* $F_{\mathcal{A}}$: $\mathcal{H}_X := \left\{ F_{\mathcal{A}}(X^T) : \mathcal{A} = (A_1, \ldots, A_L), \|A_i\|_\sigma \le s_i, \|A_i^\top - M_i^\top\|_{2,1} \le b_i \right\}$, *where each matrix has dimension at most* $W$ *along each axis. Then for any* $\epsilon > 0$,

$$\ln \mathcal{N}(\mathcal{H}_X, \epsilon, \|\cdot\|_2) \le \frac{\|X\|_2^2 \ln(2W^2)}{\epsilon^2} \left( \prod_{j=1}^{L} s_j^2 \rho_j^2 \right) \left( \sum_{i=1}^{L} \left( \frac{b_i}{s_i} \right)^{2/3} \right)^3.$$

What remains is **(III)**: Theorem 3.3 can be combined with the standard Dudley entropy integral upper bound on Rademacher complexity (see e.g. Mohri et al. [2012]), which combined with Lemma 3.1 gives Theorem 1.1.

### 3.3 Rademacher complexity lower bounds

By reduction to the linear case (i.e., removing all nonlinearities), it is easy to provide a lower bound on the Rademacher complexity of the networks studied here. Unfortunately, this bound only scales with the product of spectral norms, and not the other terms in $R_{\mathcal{A}}$ (cf. eq. (1.2)).

**Theorem 3.4.** *Consider the setting of Theorem 3.3, but all nonlinearities are the ReLU* $z \mapsto \max\{0, z\}$*, the output dimension is* $d_L = 1$*, and all non-output dimensions are at least 2 (and hence* $W \ge 2$*). Let data* $S := (x_1, \ldots, x_n)$ *be collected into data matrix* $X \in \mathbb{R}^{n \times d}$*. Then there is a* $c$ *such that for any scalar* $r > 0$, $\Re\left( \left\{ F_{\mathcal{A}} : \mathcal{A} = (A_1, \ldots, A_L), \prod_i \|A_i\|_\sigma \le r \right\}_{|S} \right) \ge c\|X\|_2 r$.

Note that, due to the nonlinearity, the lower bound should indeed depend on $\prod_i \|A_i\|_\sigma$ and not $\|\prod_i A_i\|_\sigma$; as a simple sanity check, there exist networks for which the latter quantity is 0, but the network does not compute the zero function.

### 3.4 Related work

To close this section on proofs, it is a good time to summarize connections to existing literature.

The algorithmic idea of large margin classifiers was introduced in the linear case by Vapnik [1982] (see also [Boser et al., 1992, Cortes and Vapnik, 1995]). Vapnik [1995] gave an intuitive explanation of the performance of these methods based on a sample-dependent VC-dimension calculation, but without generalization bounds. The first rigorous generalization bounds for large margin linear classifiers [Shawe-Taylor et al., 1998] required a scale-sensitive complexity analysis of real-valued function classes. At the same time, a large margins analysis was developed for two-layer networks [Bartlett, 1996], indeed with a proof technique that inspired the layer-wise induction used to prove Theorem 1.1 in the present work. Margin theory was quickly extended to many other settings (see for instance the survey by Boucheron et al. [2005]), one major success being an explanation of the generalization ability of boosting methods, which exhibit an explicit growth in the size of the function class over time, but a stable excess risk [Schapire et al., 1997]. The contribution of the present work is to provide a margin bound (and corresponding Rademacher analysis) that can be adapted to various operator norms at each layer. Additionally, the present work operates in the multiclass setting, and avoids an explicit dependence on the number of classes $k$, which seems to appear in prior work [Zhang, 2004, Tewari and Bartlett, 2007].

There are numerous generalization bounds for neural networks, including VC-dimension and fat-shattering bounds (many of these can be found in [Anthony and Bartlett, 1999]). Scale-sensitive analysis of neural networks started with [Bartlett, 1996], which can be interpreted in the present setting as utilizing data norm $\|\cdot\|_\infty$ and operator norm $\|\cdot\|_{\infty \to \infty}$ (equivalently, the norm $\|A_i^\top\|_{1,\infty}$ on weight matrix $A_i$). This analysis can be adapted to give a Rademacher complexity analysis [Bartlett and Mendelson, 2002], and has been adapted to other norms [Neyshabur et al., 2015], although the $\|\cdot\|_\infty$ setting appears to be necessary to avoid extra combinatorial factors. More work is still needed to develop complexity analyses that have matching upper and lower bounds, and also to determine which norms are well-adapted to neural networks as used in practice.

The present analysis utilizes covering numbers, and is most closely connected to earlier covering number bounds [Anthony and Bartlett, 1999, Chapter 12], themselves based on the earlier fat-shattering analysis [Bartlett, 1996], however the technique here of pushing an empirical cover through

layers is akin to VC dimension proofs for neural networks [Anthony and Bartlett, 1999]. The use of Maurey's sparsification lemma was inspired by linear predictor covering number bounds [Zhang, 2002].

**Comparison to preprint.** The original preprint of this paper [Bartlett et al., 2017] featured a slightly different version of the spectral complexity $R_{\mathcal{A}}$, given by $\left( \prod_{i=1}^{L} \rho_i \|A_i\|_\sigma \right) \left( \sum_{i=1}^{L} \frac{\|A_i - M_i\|_1^{2/3}}{\|A_i\|_\sigma^{2/3}} \right)^{3/2}$. In the present version (1.2), each $\|A_i - M_i\|_1$ term is replaced by $\|A_i^\top - M_i^\top\|_{2,1}$. This is a strict improvement since for any matrix $A \in \mathbb{R}^{d \times m}$ one has $\|A\|_{2,1} \leq \|A\|_1$, and in general the gap between these two norms can be as large as $\sqrt{d}$.

On a related note, all of the figures in this paper use the $\ell_1$ norm in the spectral complexity $R_{\mathcal{A}}$ instead of the $(2,1)$ norm. Variants of the experiments described in Section 2 were carried out using each of the $l_1$, $(2,1)$, and $l_2$ norms in the $(\sum_{i=1}^{L}(\cdot)^{2/3})^{3/2}$ term with negligible difference in the results.

Since spectrally-normalized margin bounds were first proposed in the preprint [Bartlett et al., 2017], subsequent works [Neyshabur et al., 2017, Neyshabur, 2017] re-derived a similar spectrally-normalized bound using the PAC-Bayes framework. Specifically, these works showed that $R_{\mathcal{A}}$ may be replaced by (up to $\log(W)$ factors): $\left( \prod_{i=1}^{L} \rho_i \|A_i\|_\sigma \right) \cdot L \left( \sum_{i=1}^{L} \frac{(\sqrt{W}\|A_i - M_i\|_2)^2}{\|A_i\|_\sigma^2} \right)^{1/2}$. Unfortunately, this bound never improves on Theorem 1.1, and indeed can be derived from it as follows. First, the dependence on the individual matrices $A_i$ in the second term of this bound can be obtained from Theorem 1.1 because for any $A \in \mathbb{R}^{d \times m}$ it holds that $\|A^\top\|_{2,1} \leq \sqrt{d}\|A\|_2$. Second, the functional form $(\sum_{i=1}^{L}(\cdot)^{2/3})^{3/2}$ appearing in Theorem 1.1 may be replaced by the form $L(\sum_{i=1}^{L}(\cdot)^2)^{1/2}$ appearing above by using that $\|\alpha\|_{2/3} \leq L\|\alpha\|_2$ for any $\alpha \in \mathbb{R}^L$ (this inequality following, for instance, from Jensen's inequality).

## 4 Further observations and open problems

**Adversarial examples.** Adversarial examples are a phenomenon where the neural network predictions can be altered by adding seemingly imperceptible noise to an input [Goodfellow et al., 2014]. This phenomenon can be connected to margins as follows. The margin is nothing more than the distance an input must traverse before its label is flipped; consequently, low margin points are more susceptible to adversarial noise than high margin points. Concretely, taking the 100 lowest margin inputs from `cifar10` and adding uniform noise at scale $0.15$ yielded flipped labels on $5.86\%$ of the images, whereas the same level of noise on high margin points yielded $0.04\%$ flipped labels. Can the bounds here suggest a way to defend against adversarial examples?

**Regularization.** It was observed in [Zhang et al., 2017] that explicit regularization contributes little to the generalization performance of neural networks. In the margin framework, standard weight decay ($l_2$) regularization seemed to have little impact on margin distributions in Section 2. On the other hand, in the boosting literature, special types of regularization were developed to maximize margins [Shalev-Shwartz and Singer, 2008]; perhaps a similar development can be performed here?

**SGD.** The present analysis applies to predictors that have large margins; what is missing is an analysis verifying that SGD applied to standard neural networks returns large margin predictors! Indeed, perhaps SGD returns not simply large margin predictors, but predictors that are well-behaved in a variety of other ways that can be directly translated into refined generalization bounds.

**Improvements to Theorem 1.1.** There are several directions in which Theorem 1.1 might be improved. Can a better choice of layer geometries (norms) yield better bounds on practical networks? Can the nonlinearities' worst-case Lipschitz constant be replaced with an (empirically) averaged quantity? Alternatively, can better lower bounds rule out these directions?

**Rademacher vs. covering.** Is it possible to prove Theorem 1.1 solely via Rademacher complexity, with no invocation of covering numbers?

## Acknowledgements

The authors thank Srinadh Bhojanapalli, Ryan Jian, Behnam Neyshabur, Maxim Raginsky, Andrew J. Risteski, and Belinda Tzen for useful conversations and feedback. The authors thank Ben Recht for giving a provocative lecture at the Simons Institute, stressing the need for understanding of

both generalization and optimization of neural networks. M.T. and D.F. acknowledge the use of a GPU machine provided by Karthik Sridharan and made possible by an NVIDIA GPU grant. D.F. acknowledges the support of the NDSEG fellowship. P.B. gratefully acknowledges the support of the NSF through grant IIS-1619362 and of the Australian Research Council through an Australian Laureate Fellowship (FL110100281) and through the ARC Centre of Excellence for Mathematical and Statistical Frontiers. The authors thank the Simons Institute for the Theory of Computing Spring 2017 program on the Foundations of Machine Learning. Lastly, the authors are grateful to La Burrita (both the north and the south Berkeley campus locations) for upholding the glorious tradition of the California Burrito.

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
