[Supplementary Material · supplementary.pdf]

# A Proofs

This appendix collects various proofs omitted from the main text.

## A.1 Lipschitz properties of ReLU and max-pooling nonlinearities

The standard *ReLU* ("Rectified Linear Unit") is the univariate mapping

$$\sigma_{\mathrm{r}}(r) := \max\{0, r\}.$$

When applied to a vector or a matrix, it operates coordinate-wise. While the ReLU is currently the most popular choice of univariate nonlinearity, another common choice is the *sigmoid* $r \mapsto 1/(1 + \exp(-r))$. More generally, these univariate nonlinearities are Lipschitz, and this carries over to their vector and matrix forms as follows.

**Lemma A.1.** *If $\sigma : \mathbb{R}^d \to \mathbb{R}^d$ is $\rho$-Lipschitz along every coordinate, then it is $\rho$-Lipschitz according to $\| \cdot \|_p$ for any $p \geq 1$.*

*Proof.* for any $z, z' \in \mathbb{R}^d$,

$$\|\sigma(z) - \sigma(z')\|_p = \left( \sum_i |\sigma(z)_i - \sigma(z')_i|^p \right)^{1/p} \leq \left( \sum_i \rho^p |z_i - z_i'|^p \right)^{1/p} = \rho \|z - z'\|_p.$$

$\square$

Define a *max-pooling operator* $\mathcal{P}$ as follows. Given an input and output pair of finite-dimensional vector spaces $\mathcal{T}$ and $\mathcal{T}'$ (possibly arranged as matrices or tensors), the max-pooling operator iterates over a collection of sets of indices $\mathcal{Z}$ (whose cardinality is equal to the dimension of $\mathcal{T}'$), and for each element of $Z_i \in \mathcal{Z}$ sets the corresponding coordinate $i$ in the output to the maximum entry of the input over $Z_i$: given $T \in \mathcal{T}$,

$$\mathcal{P}(T)_i := \max_{j \in Z_i} T_j.$$

The following Lipschitz constant of pooling operators will depend on the number of times each coordinate is accessed across elements of $\mathcal{Z}$; when this operator is used in computer vision, the number of times is typically a small constant, for instance 5 or 9 [Krizhevsky et al., 2012].

**Lemma A.2.** *Suppose that each coordinate $j$ of the input appears in at most $m$ elements of the collection $\mathcal{Z}$. Then the max-pooling operator $\mathcal{P}$ is $m^{1/p}$-Lipschitz wrt $\| \cdot \|_p$ for any $p \geq 1$. In particular, the max-pooling operator is 1-Lipschitz whenever $\mathcal{Z}$ forms a partition.*

*Proof.* Let $T, T' \in \mathcal{T}$ be given. First consider any fixed set of indices $Z \in \mathcal{Z}$, and suppose without loss of generality that $\mathcal{P}(T)_Z = \max_{j \in Z} T_j \geq \max_{j \in Z} T_j'$. Then

$$|\mathcal{P}(T)_Z - \mathcal{P}(T')_Z|^p = \left( \min_{j' \in Z} \max_{j \in Z} T_j - T_{j'}' \right)^p \leq \max_{j \in Z} \left( T_j - T_j' \right)^p \leq \sum_{j \in Z} \left| T_j - T_j' \right|^p.$$

Consequently,

$$\|\mathcal{P}(T) - \mathcal{P}(T')\|_p = \left( \sum_i |\mathcal{P}(T)_i - \mathcal{P}(T')_i|^p \right)^{1/p} = \left( \sum_{Z \in \mathcal{Z}} |\mathcal{P}(T)_Z - \mathcal{P}(T')_Z|^p \right)^{1/p}$$

$$\leq \left( \sum_{Z \in \mathcal{Z}} \sum_{j \in Z} |T_j - T_j'|^p \right)^{1/p} = \left( \sum_j \sum_{Z \in \mathcal{Z}: j \in Z} |T_j - T_j'|^p \right)^{1/p}$$

$$\leq \left( m \sum_j |T_j - T_j'|^p \right)^{1/p} = m^{1/p} \|T - T'\|_p.$$

$\square$

## A.2 Margin properties in Section 3.1

The goal of this subsection is to prove the general margin bound in Lemma 3.1. To this end, it is first necessary to establish a few properties of the margin operator $\mathcal{M}(v, j) := v_j - \max_{i \neq j} v_i$ and of the ramp loss $\ell_\lambda$.

**Lemma A.3.** *For every $j$ and every $p \geq 1$, $\mathcal{M}(\cdot, j)$ is 2-Lipschitz wrt $\| \cdot \|_p$.*

*Proof.* Let $v, v', j$ be given, and suppose (without loss of generality) $\mathcal{M}(v, j) \geq \mathcal{M}(v', j)$. Choose coordinate $i \neq j$ so that $\mathcal{M}(v', j) = v'_j - v'_i$. Then

$$
\mathcal{M}(v, j) - \mathcal{M}(v', j) = \left( v_j - \max_{l \neq j} v_j \right) - \left( v'_j - v'_i \right) = v_j - v'_j + v'_i + \min_{l \neq j}(-v_l)
$$
$$
\leq \left( v_j - v'_j \right) + \left( v'_i - v_i \right) \leq 2\|v - v'\|_\infty \leq 2\|v - v'\|_p.
$$

$\square$

Next, recall the definition of the ramp loss

$$
\ell_\gamma(r) := \begin{cases} 0 & r < -\gamma, \\ 1 + r/\gamma & r \in [-\gamma, 0], \\ 1 & r > 0, \end{cases}
$$

and of the ramp risk

$$
\mathcal{R}_\gamma(f) := \mathbb{E}(\ell_\gamma(-\mathcal{M}(f(x), y))).
$$

(These quantities are standard; see for instance [Boucheron et al., 2005, Zhang, 2004, Tewari and Bartlett, 2007].)

**Lemma A.4.** *For any $f : \mathbb{R}^d \to \mathbb{R}^k$ and every $\gamma > 0$,*

$$
\Pr[\arg \max_i f(x)_i \neq y] \leq \Pr[\mathcal{M}(f(x), y) \leq 0] \leq \mathcal{R}_\gamma(f),
$$

*where the* $\arg \max$ *follows any deterministic tie-breaking strategy.*

*Proof.*

$$
\Pr[\arg \max_i f(x)_i \neq y] \leq \Pr[\max_{i \neq y} f(x)_i \geq f(x)_y]
$$
$$
= \Pr[-\mathcal{M}(f(x), y) \geq 0]
$$
$$
= \mathbb{E}\mathbb{1}[-\mathcal{M}(f(x), y) \geq 0]
$$
$$
\leq \mathbb{E}\ell_\gamma(-\mathcal{M}(f(x), y))
$$

$\square$

With these tools in place, the proof of Lemma 3.1 is straightforward.

*Proof of Lemma 3.1.* Since $\ell_\gamma$ has range $[0, 1]$, it follows by standard properties of Rademacher complexity [see, for example, Mohri et al., 2012, Theorem 3.1] that with probability at least $1 - \delta$, every $f \in \mathcal{F}$ satisfies

$$
\mathcal{R}_\gamma(f) \leq \widehat{\mathcal{R}}_\gamma(f) + 2\mathfrak{R}((\mathcal{F}_\gamma)_{|S}) + 3\sqrt{\frac{\ln(2/\delta)}{2n}}.
$$

The bound now follows by applying Lemma A.4 to the left hand side.

$\square$

## A.3 Dudley Entropy Integral

This section contains a slight variant of the standard Dudley entropy integral bound on the empirical Rademacher complexity (e.g. Mohri et al. [2012]), which is used in the proof of Theorem 1.1. The presentation here diverges from standard presentations because the data metric (as in Eq. (A.1)) is not normalized by $\sqrt{n}$. The proof itself is entirely standard however — even up to constants — and is included only for completeness.

**Lemma A.5.** *Let $\mathcal{F}$ be a real-valued function class taking values in $[0,1]$, and assume that $\mathbf{0} \in \mathcal{F}$. Then*

$$\mathfrak{R}(\mathcal{F}_{|S}) \leq \inf_{\alpha>0}\left(\frac{4\alpha}{\sqrt{n}} + \frac{12}{n}\int_{\alpha}^{\sqrt{n}}\sqrt{\log\mathcal{N}(\mathcal{F}_{|S},\varepsilon,\|\cdot\|_2)}d\varepsilon.\right)$$

*Proof.* Let $N \in \mathbb{N}$ be arbitrary and let $\varepsilon_i = \sqrt{n}2^{-(i-1)}$ for each $i \in [N]$. For each $i$ let $V_i$ denote the cover achieving $\mathcal{N}(\mathcal{F}_{|S},\varepsilon_i,\|\cdot\|_2)$, so that

$$\forall f \in \mathcal{F} \quad \exists v \in V_i \quad \left(\sum_{t=1}^{n}(f(x_t)-v_t)^2\right)^{1/2} \leq \varepsilon_i, \tag{A.1}$$

and $|V_i| = \mathcal{N}(\mathcal{F}_{|S},\varepsilon_i,\|\cdot\|_2)$. For a fixed $f \in \mathcal{F}$, let $v^i[f]$ denote the nearest element in $V_i$. Then

$$\mathbb{E}_\epsilon\sup_{f\in\mathcal{F}}\sum_{t=1}^{n}\varepsilon_i f(x_t)$$

$$= \mathbb{E}_\epsilon\sup_{f\in\mathcal{F}}\left[\sum_{t=1}^{n}\epsilon_t(f(x_t)-v_t^N[f]) + \sum_{i=1}^{N-1}\sum_{t=1}^{n}\epsilon_t(v_t^i[f]-v_t^{i+1}[f]) - \sum_{t=1}^{n}\epsilon_t v_t^1[f]\right]$$

$$\leq \mathbb{E}_\epsilon\sup_{f\in\mathcal{F}}\left[\sum_{t=1}^{n}\epsilon_t(f(x_t)-v_t^N[f])\right] + \sum_{i=1}^{N-1}\mathbb{E}_\epsilon\sup_{f\in\mathcal{F}}\left[\sum_{t=1}^{n}\epsilon_t(v_t^i[f]-v_t^{i+1}[f])\right] + \mathbb{E}_\epsilon\sup_{f\in\mathcal{F}}\left[\sum_{t=1}^{n}\epsilon_t v_t^1[f]\right].$$

For the third term, observe that it suffices to take $V_1 = \{\mathbf{0}\}$, which implies

$$\mathbb{E}_\epsilon\sup_{f\in\mathcal{F}}\left[\sum_{t=1}^{n}\epsilon_t v_t^1[f]\right] = 0.$$

The first term may be handled using Cauchy-Schwarz as follows:

$$\mathbb{E}_\epsilon\sup_{f\in\mathcal{F}}\left[\sum_{t=1}^{n}\epsilon_t(f(x_t)-v_t^N[f])\right] \leq \sqrt{\mathbb{E}_\epsilon\sum_{t=1}^{n}(\epsilon_t)^2}\sqrt{\sup_{f\in\mathcal{F}}\sum_{t=1}^{n}(f(x_t)-v_t^N[f])^2} \leq \sqrt{n}\varepsilon_N.$$

Last to take care of are the terms of the form

$$\mathbb{E}_\epsilon\sup_{f\in\mathcal{F}}\left[\sum_{t=1}^{n}\epsilon_t(v_t^i[f]-v_t^{i+1}[f])\right].$$

For each $i$, let $W_i = \{v^i[f]-v^{i+1}[f] \mid f \in \mathcal{F}\}$. Then $|W_i| \leq |V_i||V_{i+1}| \leq |V_{i+1}|^2$,

$$\mathbb{E}_\epsilon\sup_{f\in\mathcal{F}}\left[\sum_{t=1}^{n}\epsilon_t(v_t^i[f]-v_t^{i+1}[f])\right] \leq \mathbb{E}_\epsilon\sup_{w\in W_i}\left[\sum_{t=1}^{n}\epsilon_t w_t\right],$$

and furthermore

$$\sup_{w\in W_i}\sqrt{\sum_{t=1}^{n}w_t^2} = \sup_{f\in\mathcal{F}}\left\|v^i[f]-v^{i+1}[f]\right\|_2$$

$$\leq \sup_{f\in\mathcal{F}}\left\|v^i[f]-(f(x_1),\ldots,f(x_n))\right\|_2 + \sup_{f\in\mathcal{F}}\left\|(f(x_1),\ldots,f(x_n))-v^{i+1}[f]\right\|_2$$

$$\leq \varepsilon_i + \varepsilon_{i+1}$$

$$= 3\varepsilon_{i+1}.$$

With this observation, the standard Massart finite class lemma [Mohri et al., 2012] implies

$$\mathbb{E}_\epsilon \sup_{w \in W_i} \left[ \sum_{t=1}^{n} \epsilon_t w_t \right] \leq \sqrt{2 \sup_{w \in W_i} \sum_{t=1}^{n} (w_t)^2 \log|W_i|} \leq 3\sqrt{2\log|W_i|}\varepsilon_{i+1} \leq 6\sqrt{\log|V_{i+1}|}\varepsilon_{i+1}.$$

Collecting all terms, this establishes

$$\mathbb{E}_\epsilon \sup_{f \in \mathcal{F}} \sum_{t=1}^{n} \epsilon_t f(x_t) \leq \varepsilon_N \sqrt{n} + 6 \sum_{i=1}^{N-1} \varepsilon_{i+1} \sqrt{\log\mathcal{N}(\mathcal{F}_{|S}, \varepsilon_{i+1}, \|\cdot\|_2)}$$

$$\leq \varepsilon_N \sqrt{n} + 12 \sum_{i=1}^{N} (\varepsilon_i - \varepsilon_{i+1}) \sqrt{\log\mathcal{N}(\mathcal{F}_{|S}, \varepsilon_i, \|\cdot\|_2)}$$

$$\leq \varepsilon_N \sqrt{n} + 12 \int_{\varepsilon_{N+1}}^{\sqrt{n}} \sqrt{\log\mathcal{N}(\mathcal{F}_{|S}, \varepsilon, \|\cdot\|_2)} d\varepsilon.$$

Finally, select any $\alpha > 0$ and take $N$ be the largest integer with $\varepsilon_{N+1} > \alpha$. Then $\varepsilon_N = 4\varepsilon_{N+2} < 4\alpha$, and so

$$\varepsilon_N \sqrt{n} + 12 \int_{\varepsilon_{N+1}}^{\sqrt{n}} \sqrt{\log\mathcal{N}(\mathcal{F}_{|S}, \varepsilon, \|\cdot\|_2)} d\varepsilon \leq 4\alpha\sqrt{n} + 12 \int_{\alpha}^{\sqrt{n}} \sqrt{\log\mathcal{N}(\mathcal{F}_{|S}, \varepsilon, \|\cdot\|_2)} d\varepsilon.$$

$\square$

### A.4  Proof of matrix covering (Lemma 3.2)

First recall the Maurey sparsification lemma.

**Lemma A.6** (Maurey; cf. [Pisier, 1980], [Zhang, 2002, Lemma 1]). *Fix Hilbert space $\mathcal{H}$ with norm $\|\cdot\|$. Let $U \in \mathcal{H}$ be given with representation $U = \sum_{i=1}^{d} \alpha_i V_i$ where $V_i \in \mathcal{H}$ and $\alpha \in \mathbb{R}_{\geq 0}^d \setminus \{0\}$. Then for any positive integer $k$, there exists a choice of nonnegative integers $(k_1, \ldots, k_d)$, $\sum_i k_i = k$, such that*

$$\left\| U - \frac{\|\alpha\|_1}{k} \sum_{i=1}^{d} k_i V_i \right\|^2 \leq \frac{\|\alpha\|_1}{k} \sum_{i=1}^{d} \alpha_i \|V_i\|^2 \leq \frac{\|\alpha\|_1^2}{k} \max_i \|V_i\|^2.$$

*Proof.* Set $\beta := \|\alpha\|_1$ for convenience, and let $(W_1, \ldots, W_k)$ denote $k$ iid random variables where $\Pr[W_1 = \beta V_i] := \alpha_i/\beta$. Define $W := k^{-1} \sum_{i=1}^{k} W_i$, whereby

$$\mathbb{E} W = \mathbb{E} W_1 = \sum_{i=1}^{d} \beta V_i \left( \frac{\alpha_i}{\beta} \right) = U.$$

Consequently

$$\mathbb{E}\|U - W\|^2 = \frac{1}{k^2} \mathbb{E} \left\| \sum_i (U - W_i) \right\|^2 = \frac{1}{k^2} \mathbb{E} \left( \sum_i \|U - W_i\|^2 + \sum_{i \neq j} \langle U - W_i, U - W_j \rangle \right)$$

$$= \frac{1}{k} \mathbb{E}\|U - W_1\|^2 = \frac{1}{k} \left( \mathbb{E}\|W_1\|^2 - \|U\|^2 \right) \leq \frac{1}{k} \mathbb{E}\|W_1\|^2$$

$$= \frac{1}{k} \sum_{i=1}^{d} \frac{\alpha_i}{\beta} \|\beta V_i\|^2 = \frac{\beta}{k} \sum_{i=1}^{d} \alpha_i \|V_i\|^2$$

$$\leq \frac{\beta^2}{k} \max_i \|V_i\|^2.$$

To finish, by the probabilistic method, there exists integers $(j_1, \ldots, j_k) \in \{1, \ldots, d\}^k$ and an assignment $\widehat{W}_i := \beta V_{j_i}$ and $\widehat{W} := k^{-1} \sum_{i=1}^{k} \widehat{W}_i$ such that

$$\left\| U - \widehat{W} \right\|^2 \leq \mathbb{E}\|U - W\|^2.$$

The result now follows by defining integers $(k_1, \ldots, k_d)$ according to $k_i := \sum_{l=1}^{k} \mathbb{1}[j_l = i]$. $\square$

As stated, the Maurey sparsification lemma seems to only grant bounds in terms of $l_1$ norms. As developed by Zhang [2002] in the vector covering case, however, it is easy to handle other norms by rescaling the cover elements. With slightly more care, these proofs generalize to the matrix case, thus yielding the proof of Lemma 3.2.

*Proof of Lemma 3.2.* Let matrix $X \in \mathbb{R}^{n \times d}$ be given, and obtain matrix $Y \in \mathbb{R}^{n \times d}$ by rescaling the columns of $X$ to have unit $p$-norm: $Y_{:,j} := X_{:,j}/\|X_{:,j}\|_p$. Set $N := 2dm$ and $k := \lceil a^2 b^2 m^{2/r}/\epsilon^2 \rceil$ and $\bar{a} := am^{1/r}\|X\|_p$, and define

$$\{V_1, \dots, V_N\} := \left\{ gY\mathbf{e}_i\mathbf{e}_j^\top : g \in \{-1, +1\}, i \in \{1, \dots, d\}, j \in \{1, \dots, m\} \right\},$$

$$\mathcal{C} := \left\{ \frac{\bar{a}}{k} \sum_{i=1}^N k_i V_i : k_i \geq 0, \sum_{i=1}^N k_i = k \right\} = \left\{ \frac{\bar{a}}{k} \sum_{j=1}^k V_{i_j} : (i_1, \dots, i_k) \in [N]^k \right\},$$
(A.2)

where the $k_i$'s are integers. Now $p \leq 2$ combined with the definition of $V_i$ and $Y$ implies

$$\max_i \|V_i\|_2 \leq \max_i \|Y\mathbf{e}_i\|_2 = \max_i \frac{\|X\mathbf{e}_i\|_2}{\|X\mathbf{e}_i\|_p} \leq 1.$$

It will now be shown that $\mathcal{C}$ is the desired cover. Firstly, $|\mathcal{C}| \leq N^k$ by construction, namely by the final equality of eq. (A.2). Secondly, let $A$ with $\|A\|_{q,s} \leq a$ be given, and construct a cover element within $\mathcal{C}$ using the following technique, which follows the approach developed by Zhang [2002] for linear prediction in which the basic Maurey lemma is applied to non-$l_1$ balls simply by rescaling.

- Define $\alpha \in \mathbb{R}^{d \times m}$ to be a "rescaling matrix" where every element of row $j$ is equal to $\|x_j\|_p$; the purpose of $\alpha$ is to annul the rescaling of $X$ introduced by $Y$, meaning $XA = Y(\alpha \odot A)$ where "$\odot$" denotes element-wise product. Note,

$$\|\alpha\|_{p,r} = \left\| (\|\alpha_{:,1}\|_p, \dots, \|\alpha_{:,m}\|_p) \right\|_r$$

$$= \left\| \left( \left\| (\|X_{:,1}\|_p, \dots, \|X_{:,d}\|_p) \right\|_p, \dots, \left\| (\|X_{:,1}\|_p, \dots, \|X_{:,d}\|_p) \right\|_p \right) \right\|_r$$

$$= m^{1/r} \left\| (\|X_{:,1}\|_p, \dots, \|X_{:,d}\|_p) \right\|_p = m^{1/r} \left( \sum_{j=1}^d \|X_{:,j}\|_p^p \right)^{1/p}$$

$$= m^{1/r} \left( \sum_{j=1}^d \sum_{i=1}^n X_{i,j}^p \right)^{1/p} = m^{1/r} \|X\|_p.$$

- Define $B := \alpha \odot A$, whereby using conjugacy of $\|\cdot\|_{p,r}$ and $\|\cdot\|_{q,s}$ gives

$$\|B\|_1 \leq \langle \alpha, |A| \rangle \leq \|\alpha\|_{p,r}\|A\|_{q,s} \leq m^{1/r}\|X\|_p a = \bar{a}.$$

  Consequently, $XA$ is equal to

$$YB = Y \sum_{i=1}^d \sum_{j=1}^m B_{ij}\mathbf{e}_i\mathbf{e}_j^\top = \|B\|_1 \sum_{i=1}^d \sum_{j=1}^m \frac{B_{ij}}{\|B\|_1} \left( Y\mathbf{e}_i\mathbf{e}_j^\top \right) \in \bar{a} \cdot \text{conv}(\{V_1, \dots, V_N\}),$$

  where $\text{conv}(\{V_1, \dots, V_N\})$ is the convex hull of $\{V_1, \dots, V_N\}$.

- Combining the preceding constructions with Lemma A.6, there exist nonnegative integers $(k_1, \dots, k_N)$ with $\sum_i k_i = k$ with

$$\left\| XA - \frac{\bar{a}}{k} \sum_{i=1}^N k_i V_i \right\|_2^2 = \left\| YB - \frac{\bar{a}}{k} \sum_{i=1}^N k_i V_i \right\|_2^2 \leq \frac{\bar{a}^2}{k} \max_i \|V_i\|_2 \leq \frac{a^2 m^{2/r}\|X\|_p^2}{k} \leq \epsilon^2.$$

  The desired cover element is thus $\frac{\bar{a}}{k} \sum_i k_i V_i \in \mathcal{C}$.

$\square$

## A.5 A whole-network covering bound for general norms

As stated in the text, the construction of a whole-network cover via induction on layers does not demand much structure from the norms placed on the weight matrices. This subsection develops this general analysis. A tantalizing direction for future work is to specialize the general bound in other ways, namely ones that are better adapted to the geometry of neural networks as encountered in practice.

The structure of the networks is the same as before; namely, given matrices $\mathcal{A} = (A_1, \ldots, A_L)$, define the mapping $F_{\mathcal{A}}$ as (1.1), and more generally for $i \leq L$ define $\mathcal{A}_1^i := (A_1, \ldots, A_i)$ and

$$F_{\mathcal{A}_1^i}(Z) := \sigma_i(A_i\sigma_{i-1}(A_{i-1}\cdots\sigma_1(A_1 Z)\cdots)),$$

with the convention $F_{\emptyset}(Z) = Z$.

- Define two sequences of vector spaces $\mathcal{V}_1, \ldots, \mathcal{V}_L$ and $\mathcal{W}_2, \ldots, \mathcal{W}_{L+1}$, where $\mathcal{V}_i$ has a norm $|\cdot|_i$ and $\mathcal{W}_i$ has norm $\||\cdot\||_i$.

- The inputs $Z \in \mathcal{V}_1$ satisfy a norm constraint $|Z|_1 \leq B$. The subscript merely indicates an index, and does not refer to any $l_1$ norm. The vector space $\mathcal{V}_1$, and moreover the collection of vector spaces $\mathcal{V}_i$ and $\mathcal{W}_i$, have no fixed meaning and are simply abstract vector spaces. However, when using these tools to prove Theorem 1.1, $\mathcal{V}_1 = \mathbb{R}^{d\times n}$ and $Z \in \mathcal{V}_1$ is formed by collecting the $n$ data points into its columns; that is, $Z = X^{\top}$.

- The linear operators $A_i : \mathcal{V}_i \to \mathcal{W}_{i+1}$ are associated with some operator norm $|A_i|_{i\to i+1} \leq c_i$:

$$|A_i|_{i\to i+1} := \sup_{|Z|_i \leq 1} \||A_i Z\||_{i+1} = c_i.$$

  As stated before, these linear operators $\mathcal{A} = (A_1, \ldots, A_L)$ vary across functions $F_{\mathcal{A}}$. When used to prove Theorem 1.1, $Z$ is a matrix (the forward image of data matrix $X^{\top}$ across layers), and these norms are all matrix norms.

- The $\rho_i$-Lipschitz mappings $\sigma_i : \mathcal{W}_{i+1} \to \mathcal{V}_{i+1}$ have $\rho_i$ measured with respect to norms $|\cdot|_{i+1}$ and $\||\cdot\||_{i+1}$: for any $z, z' \in \mathcal{W}_{i+1}$,

$$\left|\sigma_i(z) - \sigma_i(z')\right|_{i+1} \leq \rho_i \||z - z'\||_{i+1}.$$

  These Lipschitz mappings are considered fixed within $F_{\mathcal{A}}$. Note again that these operations, when applied to prove Theorem 1.1, operate on matrices that represent the forward images of all data points together. Lipschitz properties of the standard coordinate-wise ReLU and max-pooling operators can be found in Appendix A.1.

**Lemma A.7.** *Let $(\epsilon_1, \ldots, \epsilon_L)$ be given, along with fixed Lipschitz mappings $(\sigma_1, \ldots, \sigma_L)$ (where $\sigma_i$ is $\rho_i$-Lipschitz), and operator norm bounds $(c_1, \ldots, c_L)$. Suppose the matrices $\mathcal{A} = (A_1, \ldots, A_L)$ lie within $\mathcal{B}_1 \times \cdots \times \mathcal{B}_L$ where $\mathcal{B}_i$ are arbitrary classes with the property that each $A_i \in \mathcal{B}_i$ has $|A_i|_{i\to i+1} \leq c_i$. Lastly, let data $Z$ be given with $|Z|_1 \leq B$. Then, letting $\tau := \sum_{j\leq L} \epsilon_j\rho_j \prod_{l=j+1}^{L} \rho_l c_l$, the neural net images $\mathcal{H}_Z := \{F_{\mathcal{A}}(Z) : \mathcal{A} \in \mathcal{B}_1 \times \cdots \times \mathcal{B}_L\}$ have covering number bound*

$$\mathcal{N}\left(\mathcal{H}_Z, \tau, |\cdot|_{L+1}\right) \leq \prod_{i=1}^{L} \sup_{\substack{(A_1, \ldots, A_{i-1}) \\ \forall j < i. A_j \in \mathcal{B}_j}} \mathcal{N}\left(\left\{A_i F_{(A_1, \ldots, A_{i-1})}(Z) : A_i \in \mathcal{B}_i\right\}, \epsilon_i, \||\cdot\||_{i+1}\right).$$

*Proof.* Inductively construct covers $\mathcal{F}_1, \ldots, \mathcal{F}_L$ of $\mathcal{W}_2, \ldots, \mathcal{W}_{L+1}$ as follows.

- Choose an $\epsilon_1$-cover $\mathcal{F}_1$ of $\{A_1 Z : A_1 \in \mathcal{B}_1\}$, thus

$$|\mathcal{F}_1| \leq \mathcal{N}(\{A_1 Z : A_1 \in \mathcal{B}_1\}, \epsilon_1, \||\cdot\||_2) =: N_1.$$

- For every element $F \in \mathcal{F}_i$, construct an $\epsilon_{i+1}$-cover $\mathcal{G}_{i+1}(F)$ of

$$\left\{A_{i+1}\sigma_i(F) : A_{i+1} \in \mathcal{B}_{i+1}\right\}.$$

  Since the covers are proper, meaning $F = A_i F_{(A_1, \ldots, A_{i-1})}(Z)$ for some matrices $(A_1, \ldots, A_i) \in \mathcal{B}_1 \times \cdots \times \mathcal{B}_i$, it follows that

$$\left|\mathcal{G}_{i+1}(F)\right| \leq \sup_{\substack{(A_1, \ldots, A_i) \\ \forall j \leq i. A_j \in \mathcal{B}_j}} \mathcal{N}\left(\left\{A_{i+1}F_{A_1, \ldots, A_i}(Z) : A_{i+1} \in \mathcal{B}_{i+1}\right\}, \epsilon_{i+1}, \||\cdot\||_{i+2}\right) =: N_{i+1}.$$

Lastly form the cover

$$\mathcal{F}_{i+1} := \bigcup_{F \in \mathcal{F}_i} \mathcal{G}_{i+1}(F),$$

whose cardinality satisfies

$$|\mathcal{F}_{i+1}| \le |\mathcal{F}_i| \cdot N_{i+1} \le \prod_{l=1}^{i+1} N_l.$$

Define $\mathcal{F} := \{\sigma_L(F) : F \in \mathcal{F}_L\}$; by construction, $\mathcal{F}$ satisfies the desired cardinality constraint. to show that it is indeed a cover, fix any $(A_1, \dots, A_L)$ satisfying the above constraints, and for convenience define recursively the mapped elements

$$F_1 = A_1 X \in \mathcal{W}_2, \qquad G_i = \sigma_i(F_i) \in \mathcal{V}_{i+1} \qquad F_{i+1} = A_{i+1} G_i \in \mathcal{W}_{i+2}.$$

The goal is to exhibit $\widehat{G}_L \in \mathcal{F}$ satisfying $|G_L - \widehat{G}_L|_{L+1} \le \tau$. To this end, inductively construct approximating elements $(\widehat{F}_i, \widehat{G}_i)$ as follows.

- Base case: set $\widehat{G}_0 = X$.
- Choose $\widehat{F}_i \in \mathcal{F}_i$ with $\||A_i \widehat{G}_{i-1} - \widehat{F}_i\||_{i+1} \le \epsilon_i$, and set $\widehat{G}_i := \sigma_i(\widehat{F}_i)$.

To complete the proof, it will be shown inductively that

$$|G_i - \widehat{G}_i|_{i+1} \le \sum_{1 \le j \le i} \epsilon_j \rho_j \prod_{l=j+1}^{i} \rho_l c_l.$$

For the base case,

$$|G_0 - \widehat{G}_0|_1 = 0.$$

For the inductive step,

$$\begin{aligned}
|G_{i+1} - \widehat{G}_{i+1}|_{i+2} &\le \rho_{i+1} \||F_{i+1} - \widehat{F}_{i+1}\||_{i+2} \\
&\le \rho_{i+1} \||F_{i+1} - A_{i+1}\widehat{G}_i\||_{i+2} + \rho_{i+1} \||A_{i+1}\widehat{G}_i - \widehat{F}_{i+1}\||_{i+2} \\
&\le \rho_{i+1} |A_{i+1}|_{i+1 \to i+2} \left| G_i - \widehat{G}_i \right|_{i+1} + \rho_{i+1}\epsilon_{i+1} \\
&\le \rho_{i+1} c_{i+1} \left( \sum_{j \le i} \epsilon_j \rho_j \prod_{l=j+1}^{i} \rho_l c_l \right) + \rho_{i+1}\epsilon_{i+1} \\
&= \sum_{j \le i+1} \epsilon_j \rho_j \prod_{l=j+1}^{i+1} \rho_l c_l.
\end{aligned}$$

$\square$

The core of the proof rests upon inequalities which break the covering at a layer into the covering at the previous layer (handled by induction) and a cover of the present layer's weights (handled by matrix covering). These inequalities are similar to those in an existing covering number proof [Anthony and Bartlett, 1999, Chapter 12] (itself rooted in the earlier work of Bartlett [1996]); however (a) that proof operates node by node, and can not take advantage of special norms on $\mathcal{A}$, and (b) that proof does not maintain an empirical cover across layers, instead explicitly covering the parameters of all weight matrices, which incurs the number of parameters as a multiplicative factor. The idea here to push an empirical cover through layers, meanwhile, is reminiscent of VC dimension proofs for neural networks [Anthony and Bartlett, 1999, Chapter 8].

## A.6  Proof of spectral covering bound (Theorem 3.3)

The whole-network covering bound in terms of spectral and $(2,1)$ norms now follows by the general norm covering number in Lemma A.7, and the matrix covering lemma in Lemma 3.2.

*Proof of Theorem 3.3.* First dispense with the parenthetical statement regarding coordinate-wise ReLU and max-pooling operaters, which are Lipschitz by Lemmas A.1 and A.2. The rest of the proof is now a consequence of Lemma A.7 with all data norms set to the $l_2$ norm ($|\cdot|_i = \|\|\cdot\|\|_i = \|\cdot\|_2$), all operator norms set to the spectral norm ($|\cdot|_{i\to i+1} = \|\cdot\|_\sigma$), the matrix constraint sets set to $\mathcal{B}_i = \left\{ A_i : \|A_i\|_\sigma \le s_i, \|A_i^\top - M_i^\top\|_{2,1} \le b_i \right\}$, and lastly the per-layer cover resolutions $(\epsilon_1, \ldots, \epsilon_L)$ set according to

$$\epsilon_i := \frac{\alpha_i \epsilon}{\rho_i \prod_{j>i} \rho_j s_j} \qquad \text{where} \quad \alpha_i := \frac{1}{\bar{\alpha}}\left(\frac{b_i}{s_i}\right)^{2/3}, \quad \bar{\alpha} := \sum_{j=1}^L \left(\frac{b_j}{s_j}\right)^{2/3}.$$

By this choice, it follows that the final cover resolution $\tau$ provided by Lemma A.7 satisfies

$$\tau \le \sum_{j\le L} \epsilon_j \rho_j \prod_{l=j+1}^L \rho_l s_l = \sum_{j\le L} \alpha_j \epsilon = \epsilon.$$

The key technique in the remainder of the proof is to apply Lemma A.7 with the covering number estimate from Lemma 3.2, but centering the covers at $M_i$ (meaning the cover at layer $i$ is of matrices $\mathcal{B}_i$ where $A_i \in \mathcal{B}_i$ satisfies $\|A_i^\top - M_i^\top\|_{2,1} \le b_i$), and collecting $(x_1, \ldots, x_n)$ as rows of matrix $X \in \mathbb{R}^{n\times d}$. To start, the covering number estimate from Lemma A.7 can be combined with Lemma 3.2 (specifically with $p=2$, $s=1$) to give

$$\ln \mathcal{N}(\mathcal{H}_{|S}, \epsilon, \|\cdot\|_2) \tag{A.3}$$

$$\le \sum_{i=1}^L \sup_{\substack{(A_1,\ldots,A_{i-1}) \\ \forall j<i.A_j\in\mathcal{B}_j}} \ln \mathcal{N}\left(\left\{A_i F_{(A_1,\ldots,A_{i-1})}(X^\top) : A_i \in \mathcal{B}_i\right\}, \epsilon_i, \|\cdot\|_2\right)$$

$$\overset{(*)}{=} \sum_{i=1}^L \sup_{\substack{(A_1,\ldots,A_{i-1}) \\ \forall j<i.A_j\in\mathcal{B}_j}} \ln \mathcal{N}\left(\left\{F_{(A_1,\ldots,A_{i-1})}(X^\top)^\top(A_i - M_i)^\top : \|A_i^\top - M_i^\top\|_{2,1} \le b_i, \|A_i\|_\sigma \le s_i\right\}, \epsilon_i, \|\cdot\|_2\right)$$

$$\le \sum_{i=1}^L \sup_{\substack{(A_1,\ldots,A_{i-1}) \\ \forall j<i.A_j\in\mathcal{B}_j}} \ln \mathcal{N}\left(\left\{F_{(A_1,\ldots,A_{i-1})}(X^\top)^\top(A_i - M_i)^\top : \|A_i^\top - M_i^\top\|_{2,1} \le b_i\right\}, \epsilon_i, \|\cdot\|_2\right)$$

$$\le \sum_{i=1}^L \sup_{\substack{(A_1,\ldots,A_{i-1}) \\ \forall j<i.A_j\in\mathcal{B}_j}} \frac{b_i^2 \|F_{(A_1,\ldots,A_{i-1})}(X^\top)^\top\|_2^2}{\epsilon_i^2} \ln(2W^2), \tag{A.4}$$

where $(*)$ follows first since $l_2$ covering a matrix and its transpose is the same, and secondly since the cover can be translated by $F_{(A_1,\ldots,A_{i-1})}(X^\top)^\top M_i^\top$ without changing its cardinality. In order to simplify this expression, note for any $(A_1, \ldots, A_{i-1})$ that

$$\|F_{(A_1,\ldots,A_{i-1})}(X^\top)^\top\|_2 = \|F_{(A_1,\ldots,A_{i-1})}(X^\top)\|_2$$
$$= \|\sigma_{i-1}(A_{i-1}F_{(A_1,\ldots,A_{i-2})}(X^\top) - \sigma_{i-1}(0)\|_2$$
$$\le \rho_{i-1}\|A_{i-1}F_{(A_1,\ldots,A_{i-2})}(X^\top) - 0\|_2$$
$$\le \rho_{i-1}\|A_{i-1}\|_\sigma\|F_{(A_1,\ldots,A_{i-2})}(X^\top)\|_2,$$

which by induction gives

$$\max_j \|F_{(A_1,\ldots,A_{i-1})}(X^\top)^\top \mathbf{e}_j\|_2 \le \|X\|_2 \prod_{j=1}^{i-1} \rho_j \|A_j\|_\sigma. \tag{A.5}$$

Combining eqs. (A.4) and (A.5), then expanding the choice of $\epsilon_i$ and collecting terms,

$$\ln \mathcal{N}(\mathcal{H}_{|S}, \epsilon, \|\cdot\|_2) \le \sum_{i=1}^{L} \sup_{\substack{(A_1, \ldots, A_{i-1}) \\ \forall j < i. A_j \in \mathcal{B}_j}} \frac{b_i^2 \|X\|_2^2 \prod_{j<i} \rho_j^2 \|A_j\|_\sigma^2}{\epsilon_i^2} \ln(2W^2)$$

$$\le \sum_{i=1}^{L} \frac{b_i^2 B^2 \prod_{j<i} \rho_j^2 s_j^2}{\epsilon_i^2} \ln(2W^2)$$

$$= \frac{B^2 \ln(2W^2) \prod_{j=1}^{L} \rho_j^2 s_j^2}{\epsilon^2} \sum_{i=1}^{L} \frac{b_i^2}{\alpha_i^2 s_i^2}$$

$$= \frac{B^2 \ln(2W^2) \prod_{j=1}^{L} \rho_j^2 s_j^2}{\epsilon^2} \left( \bar{\alpha}^3 \right).$$

$\square$

## A.7 Proof of Theorem 1.1

As an intermediate step to Theorem 1.1, a bound is first produced which has constraints on matrix and data norms provided in advance.

**Lemma A.8.** *Let fixed nonlinearities $(\sigma_1, \ldots, \sigma_L)$ and reference matrices $(M_1, \ldots, M_L)$ be given where $\sigma_i$ is $\rho_i$-Lipschitz and $\sigma_i(0) = 0$. Further let margin $\gamma > 0$, data bound $B$, spectral norm bounds $(s_i)_{i=1}^{L}$, and $l_1$ norm bounds $(b_i)_{i=1}^{L}$ be given. Then with probability at least $1 - \delta$ over an iid draw of $n$ examples $((x_i, y_i))_{i=1}^{n}$ with $\sqrt{\sum_i \|x_i\|_2^2} \le B$, every network $F_{\mathcal{A}} : \mathbb{R}^d \to \mathbb{R}^k$ whose weight matrices $\mathcal{A} = (A_1, \ldots, A_L)$ obey $\|A_i\|_\sigma \le s_i$ and $\|A_i^\top - M_i^\top\|_{2,1} \le b_i$ satisfies*

$$\Pr\left[\arg\max_j F_{\mathcal{A}}(x)_j \ne y\right] \le \widehat{\mathcal{R}}_\gamma(f) + \frac{8}{n} + \frac{72 B \ln(2W) \ln(n)}{\gamma n} \left( \prod_{i=1}^{L} s_i \rho_i \right) \left( \sum_{i=1}^{L} \frac{b_i^{2/3}}{s_i^{2/3}} \right)^{3/2} + 3\sqrt{\frac{\ln(1/\delta)}{2n}}.$$

*Proof.* Consider the class of networks $\mathcal{F}_\lambda$ obtained by affixing the ramp loss $\ell_\gamma$ and the negated margin operator $-\mathcal{M}$ to the output of the provided network class:

$$\mathcal{F}_\gamma := \left\{ (x, y) \mapsto \ell_\gamma(-\mathcal{M}(f(x), y)) : f \in \mathcal{F} \right\};$$

Since $(z, y) \mapsto \ell_\gamma(-\mathcal{M}(z, y))$ is $2/\gamma$-Lipschitz wrt $\|\cdot\|_2$ by Lemma A.3 and definition of $\ell_\gamma$, the function class $\mathcal{F}_\gamma$ still falls under the setting of Theorem 3.3, and gives

$$\ln \mathcal{N}\left((\mathcal{F}_\gamma)_{|S}, \epsilon, \|\cdot\|_2\right) \le \frac{4B^2 \ln(2W^2)}{\gamma^2 \epsilon^2} \left( \prod_{j=1}^{L} s_j^2 \rho_j^2 \right) \left( \sum_{i=1}^{L} \left( \frac{b_i}{s_i} \right)^{2/3} \right)^3 =: \frac{R}{\epsilon^2}.$$

What remains is to relate covering numbers and Rademacher complexity via a Dudley entropy integral; note that most presentations of this technique place $1/n$ inside the covering number norm, and thus the application here is the result of a tiny amount of massaging. Continuing with this in mind, the Dudley entropy integral bound on Rademacher complexity from Lemma A.5 grants

$$\Re((\mathcal{F}_\gamma)_{|S}) \le \inf_{\alpha > 0} \left( \frac{4\alpha}{\sqrt{n}} + \frac{12}{n} \int_\alpha^{\sqrt{n}} \sqrt{\frac{R}{\epsilon^2}} \, d\epsilon \right) = \inf_{\alpha > 0} \left( \frac{4\alpha}{\sqrt{n}} + \ln(\sqrt{n}/\alpha) \frac{12\sqrt{R}}{n} \right).$$

The inf is uniquely minimized at $\alpha := 3\sqrt{R/n}$, but the desired bound may be obtained by the simple choice $\alpha := 1/n$, and plugging the resulting Rademacher complexity estimate into Lemma 3.1. $\square$

The proof of Theorem 1.1 now follows by instantiating Lemma A.8 for many choices of its various parameters, and applying a union bound. There are many ways to cut up this parameter space and organize the union bound; the following lemma makes one such choice, whereby Theorem 1.1 is easily proved. A slightly better bound is possible by invoking positive homogeneity of $(\sigma_1, \ldots, \sigma_L)$ to balance the spectral norms of the matrices $(A_1, \ldots, A_L)$, however these rebalanced matrices are then used in the comparison to $(M_1, \ldots, M_L)$, which is harder to interpret when $M_i \ne 0$.

**Lemma A.9.** *Suppose the setting and notation of Theorem 1.1. With probability at least $1 - \delta$, every network $F_{\mathcal{A}} : \mathbb{R}^d \to \mathbb{R}^k$ with weight matrices $\mathcal{A} = (A_1, \ldots, A_L)$ and every $\gamma > 0$ satisfy*

$$\Pr\left[\arg\max_j F_{\mathcal{A}}(x)_j \neq y\right]$$

$$\leq \widehat{\mathcal{R}}_\gamma(F_{\mathcal{A}}) + \frac{8}{n}$$

$$+ \frac{144 \ln(n) \ln(2W)}{\gamma n} \left(\prod_i \rho_i\right) (1 + \|X\|_2) \left(\sum_{i=1}^{L} \left(\left(\frac{1}{L} + \|A_i^\top - M_i^\top\|_{2,1}\right) \prod_{j \neq i} \left(\frac{1}{L} + \|A_j\|_\sigma\right)\right)^{2/3}\right)^{3/2}$$

$$+ \sqrt{\frac{9}{2n}} \sqrt{\ln(1/\delta) + \ln(2n/\gamma) + 2\ln(2 + \|X\|_2) + 2\sum_{i=1}^{L} \ln(2 + L\|A_i^\top - M_i^\top\|_{2,1}) + 2\sum_{i=1}^{L} \ln(2 + L\|A_i\|_\sigma)}.$$

$$\tag{A.6}$$

*Proof.* Given positive integers $(\vec{j}, \vec{k}, \vec{l}) = (j_1, j_2, j_3, k_1, \ldots, k_L, l_1, \ldots, l_L)$, define a set of instances (a set of triples $(\gamma, X, \mathcal{A})$)

$$\mathcal{B}(\vec{j}, \vec{k}, \vec{l}) = \mathcal{B}(j_1, j_2, j_3, k_1, \ldots, k_L, l_1, \ldots, l_L)$$

$$:= \left\{ (\gamma, X, \mathcal{A}) \; : \; 0 < \frac{1}{\gamma} < \frac{2^{j_1}}{n}, \; \|X\|_2 < j_2, \; \|A_i^\top - M_i^\top\|_{2,1} < \frac{k_i}{L}, \; \|A_i\|_\sigma < \frac{l_i}{L} \right\}.$$

Correspondingly subdivide $\delta$ as

$$\delta(\vec{j}, \vec{k}, \vec{l}) = \delta(j_1, j_2, j_3, k_1, \ldots, k_L, l_1, \ldots, l_L)$$

$$:= \frac{\delta}{2^{j_1} \cdot j_2(j_2 + 1) \cdot k_1(k_1 + 1) \cdots k_L(k_L + 1) \cdot l_1(l_1 + 1) \cdots l_L(l_L + 1)}.$$

Fix any $(\vec{j}, \vec{k}, \vec{l})$. By Lemma A.8, with probability at least $1 - \delta(\vec{j}, \vec{k}, \vec{l})$, every $(\gamma, X, \mathcal{A}) \in \mathcal{B}(\vec{j}, \vec{k}, \vec{l})$ satisfies

$$\Pr\left[\arg\max_j F_{\mathcal{A}}(x)_i \neq y\right] \tag{A.7}$$

$$\leq \widehat{\mathcal{R}}_\gamma(f) + \frac{8}{n}$$

$$+ \underbrace{\frac{72 \cdot 2^{j_1} \cdot j_2 \ln(2W) \ln(n)}{n^2} \left(\prod_{i=1}^{L} \rho_i\right) \left(\sum_{i=1}^{L} \left(\frac{k_i}{L} \prod_{j \neq i} \frac{l_j}{L}\right)^{2/3}\right)^{3/2}}_{=: \heartsuit}$$

$$+ \underbrace{3\sqrt{\frac{\ln(1/\delta) + \ln(2^{j_1}) + 2\ln(1 + j_2) + 2\sum_{i=1}^{L}\ln(1 + k_i) + 2\sum_{i=1}^{L}\ln(1 + l_i)}{2n}}}_{=: \clubsuit}. \tag{A.8}$$

Since $\sum_{\vec{j}, \vec{k}, \vec{l}} \delta(\vec{j}, \vec{k}, \vec{l}) = \delta$, by a union bound, the preceding bound holds simultaneously over all $\mathcal{B}(\vec{j}, \vec{k}, \vec{l})$ with probability at least $1 - \delta$.

Thus, to finish the proof, discard the preceding failure event, and let an arbitrary $(\gamma, X, \mathcal{A})$ be given. Choose the smallest $(\vec{j}, \vec{k}, \vec{l})$ so that $(\gamma, X, \mathcal{A}) \in \mathcal{B}(\vec{j}, \vec{k}, \vec{l})$; by the preceding union bound, eq. (A.8) holds for this $(\vec{j}, \vec{k}, \vec{l})$. The remainder of the proof will massage eq. (A.8) into the form in the statement of Theorem 1.1.

As such, first consider the case $j_1 = 1$, meaning $\gamma < 2/n$; then

$$\Pr\left[\arg\max_j F_{\mathcal{A}}(x)_j \neq y\right] \leq 1 < \frac{1}{\gamma n},$$

where the last expression lower bounds the right hand side of eq. (A.6), thus completing the proof in the case $j_1 = 1$. Suppose henceforth that $j_1 \geq 2$ (and $\gamma \geq 2/n$).

Combining the preceding bound $j_2 \geq 2$ with the definition of $\mathcal{B}(\vec{j}, \vec{k}, \vec{l})$, the elements of $(\vec{j}, \vec{k}, \vec{l})$ satisfy

$$2^{j_1} \leq \frac{2n}{\gamma},$$
$$j_2 \leq 1 + \|X\|_2,$$
$$\forall i \, . \quad k_i \leq 1 + L\|A_i^\top - M_i^\top\|_{2,1},$$
$$\forall i \, . \quad l_i \leq 1 + L\|A_i\|_\sigma.$$

For the term $\heartsuit$, the factors with $(\vec{j}, \vec{k}, \vec{l})$ are bounded as

$$2^{j_1} \cdot j_2 \left( \sum_{i=1}^{L} \left( k_i \prod_{j \neq i} l_j \right)^{2/3} \right)^{3/2}$$

$$\leq \frac{2n}{\gamma} \left(1 + \|X\|_2\right) \left( \sum_{i=1}^{L} \left( (L^{-1} + \|A_i^\top - M_i^\top\|_{2,1}) \prod_{j \neq i} (L^{-1} + \|A_i\|_\sigma) \right)^{2/3} \right)^{3/2}.$$

For the term $\clubsuit$, the factors with $(\vec{j}, \vec{k}, \vec{l})$ are bounded as

$$\ln(2^{j_1}) + 2\ln(1 + j_2) + 2\sum_{i=1}^{L} \ln(1 + k_i) + 2\sum_{i=1}^{L} \ln(1 + l_i)$$

$$\leq \ln(2n/\gamma) + 2\ln(2 + \|X\|_2) + 2\sum_{i=1}^{L} \ln(2 + L\|A_i^\top - M_i^\top\|_{2,1}) + 2\sum_{i=1}^{L} \ln(2 + L\|A_i\|_\sigma).$$

Plugging these bounds on $\heartsuit$ and $\clubsuit$ into eq. (A.8) gives eq. (A.6). $\qquad\square$

The proof of Theorem 1.1 is now a consequence of Lemma A.9, simplifying the bound with a $\widetilde{\mathcal{O}}(\cdot)$. Before proceeding, it is useful to pin down the asymptotic notation $\widetilde{\mathcal{O}}(\cdot)$, as it is not completely standard in the multivariate case. The notation can be understood via the $\limsup$ view of $\mathcal{O}(\cdot)$; namely, $f = \widetilde{\mathcal{O}}(g)$ if there exists a constant $C$ so that any sequence $((n^{(j)}, \gamma^{(j)}, X^{(j)}, A_1^{(j)}, \ldots, A_L^{(j)}))_{j=1}^{\infty}$ with $n^{(j)} \to \infty$, $\gamma^{(j)} \to \infty$, $\|X^{(j)}\|_2 \to \infty$, $\|A_i^{(j)}\|_1 \to \infty$ satisfies

$$\limsup_{j \to \infty} \frac{f(n^{(j)}, \gamma^{(j)}, X^{(j)}, A_1^{(j)}, \ldots, A_L^{(j)})}{g(n^{(j)}, \gamma^{(j)}, X^{(j)}, A_1^{(j)}, \ldots, A_L^{(j)}) \operatorname{poly} \log(g(n^{(j)}, \gamma^{(j)}, X^{(j)}, A_1^{(j)}, \ldots, A_L^{(j)}))} \leq C.$$

*Proof of Theorem 1.1.* Let $f = f_0 + f_1 + f_2$ denote the three excess risk terms of the upper bound from Lemma A.9, and $g = g_1 + g_2$ denote the two excess risk terms of the upper bound from Theorem 1.1; as discussed above, the goal is to show that there exists a universal constant $C$ so that for any sequence of tuples $((n^{(j)}, \gamma^{(j)}, X^{(j)}, A_1^{(j)}, \ldots, A_L^{(j)}))_{j=1}^{\infty}$ increasing as above, $\limsup_{j \to \infty} f/(g \operatorname{poly} \log(g)) \leq C$.

It is immediate that $\limsup_{j \to \infty} f_0/g = 0$ and $\limsup_{j \to \infty} f_1/(g_1 \ln(g)) \leq 144$. The only trickiness arises when studying $f_2/(g_2 \ln(g))$, namely the term $\sum_i \ln(2 + L\|A_i^\top - M_i^\top\|_{2,1})$, since $g_2$ instead has the term $\ln(\sum_i \|A_i^\top - M_i^\top\|_{2,1}^{2/3})$, and the ratio of these two can scale with $L$. A solution however is to compare to $\ln(\prod_i \|A_i\|_\sigma)$, noting that $\|(A_i)^\top\|_{2,1} \leq W^{1/2}\|A_i\|_2 \leq W\|A_i\|_\sigma$:

$$\limsup_{j \to \infty} \frac{\sum_i \ln(2 + L\|(A_i^{(j)})^\top - M_i^\top\|_{2,1})}{\ln(\prod_i \|A_i^{(j)}\|_\sigma)} \leq \limsup_{j \to \infty} \frac{\sum_i \ln(2 + L\|(A_i^{(j)})^\top\|_{2,1} + L\|M_i^\top\|_{2,1})}{\sum_i \ln(\|(A_i^{(j)})^\top\|_{2,1}/W)} = 1.$$

$\qquad\square$

## A.8 Proof of lower bound (Theorem 3.4)

*Proof of Theorem 3.4.* Define

$$\mathcal{F}(r) := \left\{ A_L \sigma_{L-1}(A_{L-1} \cdots \sigma_2(A_2 \sigma_1(A_1 x))) : \prod_{i=1}^{L} \|A_i\|_\sigma \le r \right\},$$

where each $\sigma_i = \sigma$ is the ReLU and each $A_k \in \mathbb{R}^{d_k \times d_{k-1}}$, with $d_0 = d$ and $d_L = 1$, and let $S := (x_1, \ldots, x_n)$ denote the sample.

Define a new class $\mathcal{G}(r) = \{x \mapsto \langle a, x \rangle \mid \|w\|_2 \le r\}$. It will be shown that $\mathcal{G}(r) \subseteq \mathcal{F}(C \cdot r)$ for some $C > 0$, whereby the result easily follows from a standard lower bound on $\mathfrak{R}(\mathcal{G}(r)_{|S})$.

Given any linear function $x \mapsto \langle a, x \rangle$ with $\|a\|_2 \le r$, construct a network $f = A_L \sigma_{L-1}(A_{L-1} \cdots \sigma_2(A_2 \sigma_1(A_1 x)))$ as follows:

- $A_1 = (\mathbf{e}_1 - \mathbf{e}_2)a^\top$.

- $A_k = \mathbf{e}_1 \mathbf{e}_1^\top + \mathbf{e}_2 \mathbf{e}_2^\top$ for each $k \in \{2, \ldots, L-1\}$.

- $A_L = \mathbf{e}_1 - \mathbf{e}_2$.

It is now shown that $f(x) = \langle a, x \rangle$ pointwise. First, observe $\sigma(A_1 x) = (\sigma(\langle a, x \rangle), \sigma(-\langle a, x \rangle), 0, \ldots, 0)$. Since $\sigma$ is positive homogeneous, $\sigma_{L-1}(A_{L_1} \cdots \sigma_2(A_2 y) = A_{L-1} A_{L-2} \cdots A_2 y = (y_1, y_2, 0, \ldots, 0)$ for any $y$ in the non-negative orthant. Because $\sigma(A_1 x)$ lies in the non-negative orthant, this means $\sigma_{L-1}(A_{L-1} \cdots \sigma_2(A_2 \sigma_1(A_1 x))) = (\sigma(\langle a, x \rangle), \sigma(-\langle a, x \rangle), 0, \ldots, 0)$. Finally, the choice of $A_L = \mathbf{e}_1 - \mathbf{e}_2$ gives $f(x) = \sigma(\langle a, x \rangle) - \sigma(-\langle a, x \rangle) = \langle a, x \rangle$.

Observe that for all $k \in \{2, \ldots, L-1\}$, $\|A_k\|_\sigma = 1$. For the other layers, $\|A_L\|_\sigma = \|A_L\|_2 = \sqrt{2}$ and $\|A_1\|_\sigma = \sqrt{2} \cdot r$, which implies $f \in \mathcal{F}(2r)$.

Combining the pieces,

$$\mathfrak{R}(\mathcal{F}(2r)_{|S}) \ge \mathfrak{R}(\mathcal{G}(r)_{|S}) = \mathbb{E} \sup_{a : \|a\|_2 \le r} \sum_{t=1}^{n} \epsilon_t \langle a, x_t \rangle = r \cdot \mathbb{E} \left\| \sum_{t=1}^{n} \epsilon_t x_t \right\|_2.$$

Finally, by the Khintchine-Kahane inequality there exists $c > 0$ such that

$$\mathbb{E} \left\| \sum_{t=1}^{n} \epsilon_t x_t \right\|_2 \ge c \cdot \sqrt{\sum_{t=1}^{n} \|x_t\|_2^2} = c\|X\|_2. \qquad \square$$