[Reviews · NeurIPS 2017]

Reviewer 1



Updates after rebuttal: some of my concerns are addressed, so I updated the rating a bit. However, the last point, quoting the rebuttal "Spectral complexity can be compared across architectures, and indeed part of our aim in obtaining a bound with no explicit scaling in the number of layers was to compare architectures with drastically different numbers of layers." What I meant is actual experiments that apply the proposed technique to empirically compare networks with very different architectures. It would be great if at least one such experiment could be included in the final version if this paper get accepted. ------------------------ This paper investigate the question of measuring the effective complexity or generalization capability of a network and propose to do it via the notion of margin normalized by a spectral complexity measure. With a covering number based analysis, a margin based generalization bound is derived to suggest that, if the training could be achieved with large margin and bounded R_A, then the test accuracy can be controlled effectively. Based on the theoretical results, the normalized margin (by the spectral complexity measure) is proposed as a measure on various networks. Among with other results, it is shown that this metric could effectively distinguish the networks trained on random labels and true labels (while the unnormalized margin could not). Although a lot of open questions remains (as stated in the end of the paper), the current paper is already quite interesting. The reviewer is willing to change the rating if the following technical details are addressed: - Please add ticks to all the plots. Currently it is very hard to interpret without knowing the scale of the axis. For example, in Figure 1, is the mean of the two margin distributions further apart in (c) than in (b)? - Could you also plot the mean of the normalized vs unnormalized margins as training goes (i.e. like Figure 1(a)) on cifar and rand label? - Could you add caption to Figure 3? - What is W in Theorem 1.1? - F_A in line 63 is a typo? - Could this measure be used to compare across architectures? Different architectures could have different test accuracy on cifar. Could you take a number of different architectures and compare their test accuracy vs. their normalized margins?

Reviewer 2



This paper proposes analysing the margin distributions of deep neural network to analyse their generalization properties. In particular they propose a normalized margin estimator, which is shown to reflect generalization properties of certain neural networks much better than the naive margin computation. Your notion of margin is different than what I'm used to. (I usually hear margin defined as the minimum distance in input space to a point of a different class, as SVMs are commonly depicted). Maybe just take a few words to stress this? Why does the x axis on some (if not all?) your figures not start at 0? Shouldn't the margin always be positive? Unless you're not taking y to be the argmax of F(x) but instead the true class, in which case it would be negative for a missclassification. Maybe mention this. What is the data distribution in your plots? Train, validation, or test? I general your plots should have clearer axes or captions. In Theorem 1.1 it's not clear to me what B and W are (without the appendix), maybe just briefly mention what they are. CIFAR and MNIST are acronyms, and should be capitalized. Ditto for SGD. There are some typos throughout, and I feel like the writing is slightly "casual" and could be improved. Your section on adversarial examples is somewhat puzzling to me. The margins, let's say M1, that you describe initially are in Y-space, they're the distance between the prediction f(x)_y and the next bess guess max_i!=y f(x)_i. This is different than the input space margin, let's say M2, which is something like `min_u ||u-x|| s.t. f(u) \neq f(x)`. As such when you say "The margin is nothing more than the distance an input must traverse before its label is flipped", you're clearly referring to M2, while it would seem your contribution revolves around M1 concept of margin. While intuitively M1 and M2 are probably related, unless I'm missing something about your work relating them, claiming "low margin points are more susceptible to adversarial noise than high margin points" is a much stronger statement than you may realise. Overall I quite like this paper, both in terms of the questions it is trying to answer, and in the way in does. On the other hand, the presentation could be better, and I'm still not convinced that the margin in output space has too much value in the way we currently train neural networks. We know for example that the output probabilities of neural networks are not predictive of confidence (even to an extent for Bayesian neural networks). I think more work relating the input margin with the output margin might be more revealing. I've given myself a 2/5 confidence because I am not super familiar with the math used here, so I may be failing to get deeper implications. Hopefully I'll have time to go through this soon.